# Human Milk Oligosaccharides as Potential Antibiofilm Agents: A Review

**DOI:** 10.3390/nu14235112

**Published:** 2022-12-01

**Authors:** Ankurita Bhowmik, Phatchada Chunhavacharatorn, Sharanya Bhargav, Akshit Malhotra, Akalya Sendrayakannan, Prashant S. Kharkar, Nilesh Prakash Nirmal, Ashwini Chauhan

**Affiliations:** 1Department of Microbiology, Tripura University, Agartala 799022, India; 2Institute of Nutrition, Mahidol University, Salaya, Nakhon Pathom 73170, Thailand; 3Department of Molecular Biology, Yuvaraja’s College, Mysuru 570005, India; 4Invisiobiome, New Delhi 110066, India; 5Department of Food Engineering and Technology, Institute of Chemical Technology (ICT), Nathalal Parekh Marg, Matunga, Mumbai 400019, India; 6Department of Pharmaceutical Sciences and Technology, Institute of Chemical Technology (ICT), Nathalal Parekh Marg, Matunga, Mumbai 400019, India

**Keywords:** human milk oligosaccharides, biofilm, gut microbiota, antimicrobial resistance

## Abstract

Surface-associated bacterial communities called biofilms are ubiquitous in nature. Biofilms are detrimental in medical settings due to their high tolerance to antibiotics and may alter the final pathophysiological outcome of many healthcare-related infections. Several innovative prophylactic and therapeutic strategies targeting specific mechanisms and/or pathways have been discovered and exploited in the clinic. One such emerging and original approach to dealing with biofilms is the use of human milk oligosaccharides (HMOs), which are the third most abundant solid component in human milk after lactose and lipids. HMOs are safe to consume (GRAS status) and act as prebiotics by inducing the growth and colonization of gut microbiota, in addition to strengthening the intestinal epithelial barrier, thereby protecting from pathogens. Moreover, HMOs can disrupt biofilm formation and inhibit the growth of specific microbes. In the present review, we summarize the potential of HMOs as antibacterial and antibiofilm agents and, hence, propose further investigations on using HMOs for new-age therapeutic interventions.

## 1. Introduction

Bacteria possess a unique capability to form biofilms that are ubiquitous in nature. It is a multistage and elaborate process that begins with bacterial adhesion to surfaces, followed by the synthesis of extracellular polymeric substance (EPS) matrix, development of microcolonies, and finally concludes with the dispersion of the bacterial cells from the initial [1,2]. The morphology of biofilms can be diverse and that mostly relies on the integral bacterial species and the circumstances under which the biofilm was originally formed [3,4]. The development of biofilms protects the bacteria against enzymatic degradation, antimicrobials, and host defense systems [5,6]. Population heterogeneity, slow metabolic activity, increased efflux pumps, and presence of persister subpopulations are some of the major factors that reduce antimicrobial susceptibility in biofilms. Studies show multifold tolerance of biofilms towards antibiotics compared with planktonic cells that, apart from impairing the treatment efficacy, also act as a resistant adaptive mechanism for bacterial survival [7]. In clinical settings, more than 80% of all nosocomial infections are of biofilm origin and they pose a crucial threat to human health [8,9,10]. Several biofilm-associated infections are listed in Table 1.

The reduced antibiotic susceptibility of the pathogens in biofilms urgently calls for alternative treatments to manage biofilm-associated tissue and device infections. In order to tackle the menace of multidrug-resistant biofilms, a diverse and synergistic approach concerning the contributing factors of biofilm genesis and establishment of antimicrobial resistance (AMR) is of utmost importance. Lately, the use of chelators, plant extracts, natural products, nanoparticle coatings, quorum quenchers, and a few others have proved their potential as preventive strategies against the biofilms of clinically relevant bacteria using either *in vitro* or *in vivo* models [8,9,19]. Because biofilm development is a multistage process, targeting its initial stages can potentially prevent biofilm formation and further progression of pathogenic events.

Non-digestible oligosaccharides (NDOs) hold high potential in preventing biofilm formation. They are promising antibacterial and biofilm inhibitory compounds and can be used as adjuvants with antibiotics in augmentation therapy [20]. They are diverse, mostly differing in monosaccharide constituents. These include fructo-oligosaccharides (FOS), galacto-oligosaccharides (GOS), alginate oligosaccharides (AOS), chitosan oligosaccharides (COS), pectic oligosaccharides (POS), xylo-oligosaccharides, etc. [21]. Among these, FOS and GOS are broadly studied for their prebiotic properties, as they enhance the growth of beneficial microbes [22,23,24]. NDOs are also reported for their selective anti-adhesive properties against pathogenic bacteria. They competitively interfere with the recognition process of host cells by pathogenic bacteria due to their structural similarities with the surface proteins present on host cells [25]. Furthermore, COS in combination with antibiotics showed enhanced antibacterial and antibiofilm activity. For instance, COS as an adjunct with florfenicol is effective against resistant swine *streptococcus* [26]. The adjuvant therapy involving COS and clindamycin effectively inhibited an *S.aureus* biofilm [20]. COS and AOS also showed biofilm inhibitory and anti-virulence properties against *Acinetobacter baumannii* and *Pseudomonas aeruginosa* [27]. AOS alone and in combination with ampicillin was considered to be very efficient against *E. coli* [28]. Moreover, AOS that released nitric oxide was effective in inhibiting biofilm formation in methicillin-resistant *S. aureus* [28]. Over the last few decades, NDOs have been introduced in food industries as a functional ingredient in milk products, beverages, desserts, confectionary products, meat products, and probiotic-based food items. They have also found applications in pharmaceutical industries, such as bulking agents, cosmetic stabilizers, immunostimulating agents, etc. [29]. However, the safety consideration for the use of NDOs is still in question. For instance, FOS administration above 20 g/day is reported to have certain side effects such as bloating, abdominal cramps, diarrhea, etc. Moreover, preclinical studies demonstrate the neurodevelopmental properties of NDOs, which turn out to be detrimental at clinical levels [30]. Thus, more reproducible studies on the toxicity and tolerability of NDOs are required to validate their potential efficacy as prebiotics.

On the contrary, human milk oligosaccharides (HMOs) are more suitable and safer for human consumption and they are already used in infant food, e.g., CARE4U™ [31,32,33]. They are unconjugated polysaccharides present in breast milk that forms a distinctive class of complex NDOs and are the first set of prebiotics that are degraded by the intestinal microbiota [34]. They are the third most abundant constituent of human milk [35] after lactose, constituting approximately 5–15 g/L of mature breast milk. The amount and type of HMOs present in human milk depend upon several factors including the genetic aspects, food habits, nutrient intake, and blood glucose levels of the lactating mother and the lactating period [36,37,38]. It was back in the 20th century that the discovery and importance of HMOs in preventing infectious diseases came to light. Moro, Escherich, and Tissier were among the first researchers who observed the distinctiveness in the microbiota composition between bottle-fed and breast-fed infants [36]. Moro concluded that breast milk contains certain “growth factors” that stimulate intestinal flora in infants. After 50 years, in 1954 “the growth factor” postulated by Moro was identified by Paul Gyorgy and Richard Kuhn as N-acetylglucosamine (GlcNAc) [32]. To date, more than 200 structurally distinct HMOs have been identified. The HMOs are tolerant of the acidic conditions of the stomach and hydrolysis by the digestive enzymes [37]. Undigested HMOs are conveyed to the large intestine where they are fermented by the gut microbes that stimulate the growth and development of commensal bacteria, especially *Lactobacilli* and *Bifidobacteria* [33]. HMOs also constitute part of the nonspecific immune response and could act as immunomodulators [38]. Interestingly, few HMOs can act on proinflammatory cytokine secretion and promote inflammatory cytokines synthesis. They can prevent bacterial colonization by establishing an innate immune response leading to the activation of chemokines or cytokines including interleukin (IL)-1β, IL-8, and IL-17C [39]. The present review intends to cover the comprehensive understanding of antibacterial as well as antibiofilm characteristics of HMOs. The intricate interplay between the structures of HMOs and their therapeutic activities holds great potential in the development of a new array of interventions to combat dreadful biofilm infections.

## 2. Biofilm Formation

The intense involvement of bacterial biofilms in chronic infections has always drawn the much-needed attention of the scientific community to understand its development and antimicrobial resistance mechanisms. Biofilm development in bacteria is dependent on specific environmental factors where they transition themselves from a free-floating planktonic to a surface-attached sessile form (Figure 1). In general, the degree of adhesion of bacteria on biotic or abiotic surfaces is determined by several forces such as Brownian motion, van der Waals forces, hydrodynamic interactions, etc. [40]. In *Staphylococci*, surface proteins, such as SasX, FnBPA, FnBPB, Bap, etc., collectively termed microbial surface components, recognizing adhesive matrix molecules (MSCRAMMs) are involved in the initial attachment [41]. In *P. aeruginosa,* the surface attachment is mediated by type-IV pili using twitching movement, the absence of which results in aberrant biofilm formation [42]. The WspA protein of *P. aeruginosa* recognizes substrate receptors upon surface contact and initiates c-di-GMP synthesis. At a high concentration, c-di-GMP promotes CdrA and cup fimbrial adhesins production, which along with other components of a biofilm matrix such as B-band LPS, Psl exopolysaccharides, and eDNA enhance surface adherence [43].

Once adhered, the bacterial cells secrete EPS that encases the cells and facilitates multilayered biofilm formation. The biofilm matrix consists of exopolysaccharides, proteins, nucleic acids, and other polymers that strengthen the biofilm structure as well as provide protection against antibiotic stress and the host immune response [44]. A study carried out on *K. pneumonia* showed that lactamase enzymes present in the biofilm matrix of the wild strain inhibited the penetration of ampicillin, whereas its absence in the mutant strain could not prevent its filtration [45]. As the cells proliferate and develop microcolonies, the biofilm gradually shapes into a 3-dimensional, mushroom-like structure equipped with small channels for transporting nutrients, water, and waste at different layers of the biofilm matrix [5]. Depending on the physicochemical conditions prevailing in different parts of the biofilm, the cell density along with their gene expression varies promptly. The outermost regions retain the metabolically active cells, while at the core, the cells are largely nongrowing and are in the dormant phase. Among these dormant cells, the persister population possesses high tolerance toward antibiotics and is extremely difficult to eradicate [46]. Eventually, the dispersion of the non-surface-attached biofilm cells takes place via signal-mediated hydrolysis of the EPS layer that spreads to a new environment. These dispersed cells, being highly virulent, are believed to cause acute infections [47].

In a biofilm, the population density, metabolic activity, and dislodging of cells are regulated by quorum sensing (QS). The QS in *S. aureus* is mediated by autoinducing peptides (AIP) regulated by the *agr* locus. Agr downregulates MSCRAMMs formation, which in turn encourages the dispersion of biofilm biomass and also upregulates toxin production. This is advantageous for bacteria in regulating acute infections. During chronic infections, the QS signaling is irreversibly inactivated, which leads to extensive biofilm growth with a loss of ability to disseminate from the surface of infection. *Staphylococci* spp. also possess a LuxS/AI-2 mediated QS system. It produces an AI-2 autoinducer that regulates virulence, capsule synthesis, antibiotic susceptibility, and biofilm formation. It is reported that LuxS controls polysaccharide intercellular adhesin (PIA)-dependent biofilm formation by repressing rbf expression [48]. 

In *P. aeruginosa,* the las QS system regulates the structure of the biofilm matrix. In lasI mutants, flat and undifferentiated biofilms were dominant compared with the wild-type strain [49]. AHL-mediated QS systems in *P. aeruginosa* control the release of eDNA and Pel polysaccharides in biofilms. QS also controls the synthesis of rhamnolipids in *P. aeruginosa* in the center of the mushroom cap of the biofilm, which favors cell dispersion by disrupting the non-covalent interactions between matrix molecules and biofilm cells.

## 3. Different Classes of HMOs and Their Receptors

The molecular structure of HMOs is the principal factor for their lectin specificity expressed on human cells and their distinctive metabolism by gut microbes. The structural variability in HMOs is attributed to monosaccharide constituents, rate of polymerization, charge, and acetylation. In general, monosaccharides including glucose, galactose, *N*-acetylglucosamine, fucose, and sialic acid make up the HMO molecules (Figure 2).

A lactose molecule forms the backbone of all HMOs [32,34]. The lactose residue is further modified by the addition of *N*-acetylglucosamine by β1–3 or β1–6 bonds. β1–3 linkages elongate the fundamental structure into linear patterns, whereas β1–6 bonds introduce branching. Two enzymes, namely fucosyltransferase and sialyltransferase, are involved in the alteration of the basic structure of HMOs. They catalyze the addition of fucose and/or sialic acid through alpha 1-2,3,4 and alpha 2-3,6 bond formation at the terminal position [50,51]. HMOs are broadly classified into three categories, fucosylated, sialylated, and neutral (Figure 3), based on the presence of sialic acid and fucose residues [52].

Neutral HMOs comprise 42–55% of the total HMOs. Lacto-N-tetraose (LNT) is an example of this class [32,53]. Lacto-N-tetraose (LNT) is an amino tetrasaccharide comprising β-D-galactose, N-acetyl-b-D-glucosamine, β-D-galactose, and D-glucose residues in a linear sequence. Lacto-N-neotetraose (LNnT) is an isomer of LNT, which is formed by the bonding of terminal β-D-galactose with an N-acetyl-β-D-glucosamine moiety via β (1–4) linkage. LNnT is the second predominant HMO in human milk [54,55].

Fucosylated HMOs constitute up to 35–50% of the total HMOs. The HMO 2′-Fucosyllactose (2′-FL) is the richest component in human milk that falls under this category [53,56]. This trisaccharide (molecular weight: 488.44 Da) consists of L-fucose bound to lactose at the second or third position.

About 12–14% of the total HMOs are acidic in nature and mostly contain sialic acid in their structure. Sialylation of oligosaccharides results in the incorporation of negatively- charged units into neutral HMOs. Examples of acidic HMO are 3′–sialyl lactose (3′-SL) and 6′-sialyl lactose [32,53]. The trisaccharide 3’-sialyllactose is composed of N-acetylneuraminic acid, β-D-galactose, and D-glucose and is formed by the linking of an acetyl neuraminyl (NANA) moiety at the third position of the β-D-galactosyl moiety of the lactose unit. The 6′-sialyllactose (6′-SL) structure is similar to 3′-SL except the bonding occurs at the sixth position between the NANA entity and the β-D-galactosyl entity of lactose [57].

The literature suggests different glycan-binding proteins, specifically known as lectins, as HMO receptors. For example, galectins, siglecs, c- type lectins, and selectins bind to specific HMO molecules to exert their functions [58,59].

Galectins are small, soluble, beta-galactoside-specific lectins with conserved carbohydrate-recognition domains (CRDs) that are expressed on intestinal epithelia, immune cells, skeletal muscle, lymphoid tissues, the heart, kidneys, neurons, dendritic cells, etc. There are three types of galectins: prototype, chimera, and tandem galectins. In the CRD of prototype galectin, eight amino acids, namely His44, Arg48, Trp68, Val59, Asn61, Asn46, Glu71, and Arg73, are responsible for glycan binding. The OH groups at C-4 and C-6 of galactose and C-3 of N-acetylglucosamine of HMOs form hydrogen bonds with the amino acid residues of galectin in the CRD region (Figure 4). In tandem galectins, two CRD regions are present at the N and C terminus, separated by a short proline and glycine-rich peptide linker. HMOs containing a Galβ1–3GalNAc residue bind to the CRD at the N terminus of tandem galectins via the OH group. As galectins are involved in converting signals to cells and regulating cell functions, the binding of HMOs with galectins controls the interactions between galectins with ligands expressed in other cells. LNnT, LNT, LNFP-II, LNFP-III, and LNDFH are the common HMOs that bind with galectins 1,2,3,7,8,9 [58,59,60].

Another class of lectins, called siglecs, is reported to bind sialylated HMOs. They are sialic acid-binding immunoglobulin-like lectins that are expressed on different blood cells such as NK cells, dendritic cells, neutrophils, macrophages, etc. All siglecs possess the V-set immunoglobulin-like domain at the N-terminal that interacts with sialic acid of glycoproteins, glycolipids, and HMOs, such as 3′SL and 6′SL.

Additionally, fucosylated and sialylated HMOs bind to other lectins such as selectins and C-type lectins. The binding of HMOs with selectins reduces selectin-mediated leukocyte–endothelial cell and leukocyte–platelet interactions. C-type lectins that are expressed on dendritic cells and macrophages play a crucial role in antigen presentation and immune response inhibition. For instance, the interaction of HMOs with C-type lectins, e.g., DC-SIGN, inhibits the transfer of HIV-1 to CD4 T cells [31]. HMOs’ interactions with these receptors thus indicate their role in regulating adaptive and innate immune responses.

## 4. Antimicrobial Properties of Human Milk Oligosaccharides (HMOs)

Recent advances in *in vitro* and *in vivo* studies have highlighted the diverse antimicrobial properties of HMOs. They serve as metabolic substrates for gut microbiota, improve gut barrier functions, act as decoy receptors in preventing pathogen adhesion on the mucosal membrane, regulate immune responses, and thereby reduce the rate of infections.

### 4.1. HMOs Shape Gut Flora and Gut Immune Function

The human gut is a vital organ needed for maintaining all the functions of the human body. Colonization of the gut by microbes plays a critical role in host metabolism, mostly established in a child’s life during 2–3 years of age. The human gut also helps in maintaining the immune, gastrointestinal, and neural systems [61]. 

In the gut, intestinal epithelial cells absorb acidic HMOs by the nonspecific paracellular route and neutral HMOs by the receptor-mediated transcellular pathway [37,62]. Studies have shown that HMOs modulate protein expression of gut epithelial cells. In an *in vivo* study, supplementation of sialylated HMOs in rats reduced the expression of genes responsible for the secretion of IL-12, L-8, NF-kB, and TNFα by activating the anti-inflammatory peptidoglycan recognition protein 3 (PglyRP3) [63,64]. It is also well established that HMOs, specifically 2′FL, inhibit the secretion of proinflammatory cytokines in HT-29 and Hep-2 cell lines that significantly reduce the invasion of *C. jejuni* [65].

Furthermore, HMOs strengthen the intestinal barrier by modulating the synthesis of mucin [66]. It was observed that HMO supplementation restores goblet cells in rats with necrotizing enterocolitis. Goblet cells synthesize Muc2 and other factors that form the protective mucous layer over the intestinal epithelial cells [67]. It was also demonstrated that 2′FL and LNnT induce the expression of claudin-5 and claudin-8 proteins, which are significant in strengthening the tight junctions and limiting the permeability of molecules through the intestinal walls [68,69].

HMO molecules enrich the gut microbiome, specifically by promoting the colonization by a *Bifidobacterium*-dominated bacterial community [70]. Among *Bifidobacteria*, a distinct class of gene cluster encoding fucosidase and sialidases enzymes and their specific transporters is required for breaking down the complex HMOs [51].

In *Bifidobacteria,* HMO molecules are degraded by either intracellular or extracellular strategies [71]. Intracellular digestion includes the transportation of HMOs inside the cell via ABC transporters and enzymatic hydrolysis of the HMO molecule into monosaccharides by glycosyl hydrolases [72]. In the extracellular process, the HMOs are degraded into their monomers outside the cell by cell-surface-associated glycosidases and the monosaccharides are transported inside the cell. The digested HMOs are then assimilated into the bacterial central metabolism pathway, which releases short chain fatty acids (SCFA) as end products [33,73]. SCFA serves as an intermediate molecule in connecting gut flora with the immune system. They modify gene expression, differentiation, and apoptosis of epithelial cells; inhibit the function of histone deacetylases; and activate G-protein coupled receptors (GPCRs). GPCRs, along with other transcriptional factors, modulate the function and development of leukocytes. All the SCFAs also possess other specific functions. For instance, acetic acid and butyric acid act as a source of energy for muscle and skeletal tissue and colonocytes, respectively. Butyric acid also improves the gut epithelial barrier, regulates the proliferation and activity of regulatory T cells, and enhances the metabolism of intestinal epithelial cells [74,75]. The SCFAs are crucial for appetite as they activate free fatty acid receptors that, in turn, elevate the circulation of intestinal anorectic hormones [76]. Additionally, they exhibit antimicrobial properties against viral, bacterial, and protozoal infections. The SCFAs, along with sphingomyelin, promote the myelination of the central nervous system [53]. Moreover, the synthesis of SCFAs reduces the pH of the colon, which is non-conducive for the survival of pathogenic bacteria [77]. 

The HMOs also promote the growth of other commensal bacteria such as *Akkermansia* and *Lactobacillus. Akkermansia* increases the secretion of mucin which reduces the colonization of harmful bacteria [78]. In 2022, an *in vivo* study by Kathryn and group demonstrated the protective activity of pooled HMOs against Group B *Streptococcus* (GBS) colonization in the vagina without altering the vaginal flora [79].

Finally, the metabolism of HMOs and the end products are selective among different bacterial strains. It is dependent on the specific gene clusters present in bacteria and the molecular structure of HMO molecules. For instance, *B. breve* and *B. longum* actively degrade LNT, whereas digestion of fucosylated HMOs is more conducive for *B. bifidium* and *B. infantis* [80,81].

### 4.2. HMOs Prevent the Growth and Colonization of Pathogenic Bacteria

As discussed above, HMOs serve as a selective substrate and specifically favor the growth of symbiotic bacteria. As a result, the gut flora outcompetes the pathogenic bacteria for space and nutrients. Additionally, the end products of HMO metabolism, such as SCFAs, lower the intestinal pH that stunts the growth and proliferation of harmful bacteria.

Furthermore, HMO molecules act as anti-adhesive molecules and prevent the colonization of pathogenic bacteria.

The first step of bacterial colonization involves the adhesion of bacteria to the host cell surface, which is mediated by the binding of bacterial surface ligands to the host cell oligosaccharide receptors. HMOs behave as decoys and bind efficiently to the pathogen adhesins due to their smaller size [82,83]. Thus, HMOs act as receptor analogs that bind to pathogens and prevent their colonization, thereby reducing the incidence of infectious diseases (Figure 5) [25]. 

Another mode of eradicating pathogens adopted by the HMOs is due to their ability to compete with the pathogens for binding to the cell surface receptors. HMOs prevent pathogens from adhering to the cell surface carbohydrate receptors by behaving as carbohydrate-binding ligands or soluble ligand analogs which compete with bacterial adhesins to bind to these receptors and thus competitively inhibit their attachment, further preventing their colonization (Figure 6) [25].

The anti-adherence properties of HMOs are reported against *Shigella* sp., *Campylobacter* sp., and various pathotypes of *E. coli* [84,85]. The HMOs also prevent the attachment of *L. monocytogenes* and *S. pneumoniae* to the host cell surface [53,86]. Several *in vivo* and *in vitro* studies have displayed compromised GBS colonization in the vagina without affecting the pre-existing vaginal microbiome. This can evolve as a promising preventive therapy against vaginosis [79]. Studies revealed that the acidic fraction of HMOs effectively alters the expression of genes responsible for the arrangement of surface receptors, which affects pathogen binding. This type of interaction was observed in the presence of 3′-SL against enteropathogenic *E. coli* [87]. The anti-adhesive properties of HMOs are influenced by the charge and molecular weight of the oligosaccharide and the targeted pathogen [35].

In the neutral HMOs, fucosylated fractions have been reported to exhibit intermediate levels of adhesion inhibition (up to 50%). Neutral HMOs exhibit antipathogenic functionalities depending on their molecular weight and pathogenic species. The high molecular weight fragments (HMWF) are known to inhibit adhesion in *E. coli* and *Vibrio cholerae*, and low molecular weight fragments (LMWF) inhibit the adhesion of *E. coli* and *Salmonella fyris.* However, HMWF failed to inhibit *S. fyris* and LMWF failed to inhibit *V. cholerae*, indicating their pathogen specificity [88]. Pathogen-inhibiting activity also depends upon the fucosylation patterns. Research demonstrated that 2′-FL has shown inhibitory activity against several pathogen species. Studies using 2′-FL against *C. jejuni* have reported the inhibition of the pathogen to human intestinal epithelial cells [65,89]. In other work, 2′-FL has also shown anti-adhesive effects against enteropathogenic *E. coli* (EPEC) and *V. cholerae* to intestinal epithelial cells (Caco-2). Additionally, 2′-FL also retards *Pseudomonas spp*. infection [56]. The anti-adhesive activity of 2′-FL was also observed against *Neisseria meningitides*, whose adhesion to salivary agglutinin was effectively inhibited. Recent studies involving the use of 2′-FL against *S. agalactiae,* and its serotypes have shown an adhesion inhibitory effect [89]. It has been found that 2′-FL competes with pathogens to bind to the cell surface glycan receptors, which prevents colonization and invasion of the pathogens. The unbound pathogen is then eliminated without causing any infections [89]. Like 2′-FL, 3-FL also exhibits anti-adhesive effects; 3-FL inhibited the adhesion of *E. coli* by 30% and *S. fyris* by 16%. It also reduced the binding of *P. aeruginosa* to the human respiratory cell line A549 by up to 23% [90]. Neutral N-containing HMOs such as LNnT are also efficient in inhibiting the adhesion of *S. pneumoniae* [88]. Furthermore, acidic HMOs also contribute towards the inhibition of pathogen adhesion. One study showed that 3′-SL, in particular, has inhibited cellular adhesion in *E. coli* serotype O119, *H. pylori,* and *V. cholerae* [88], and 3′-SL is also known to reduce the invasion of uropathogenic *E. coli* (UPEC) [91]. Similarly, 6′-SL has shown anti-adhesive effects against *S. pneumoniae.* It also inhibited the adhesion of *Salmonella fyris* to buccal epithelial cells [88]. Studies showed that the presence of acidic HMOs inhibited the expression of several fimbrial types in *E. coli.* Unlike neutral HMO fractions, sialylated oligosaccharides have reduced pathogen specificity. This is because sialylated oligosaccharides, being negatively charged, readily commute with the oppositely charged segments present on the cell surface [34]. Furthermore, HMOs are reported to avert necrotizing enterocolitis (NEC). NEC etiology can attribute to the extensive establishment of bacterial and mucosal neutrophil infiltration. *In vivo* studies revealed that the introduction of HMOs resulted in the elevated expression of muc2, resulting in the release of mucin proteins, which makes enterocytes impermeable to dextran, thereby inhibiting microbial adhesion, reducing the activation of neutrophils, and thus lowering mucosal neutrophil activity [76,92]. In addition, HMOs containing galactose and mannose monosaccharide moieties were observed to prevent adhesion in many bacterial species, which include *C. jejuni*, *Citrobacter rodentium*, *Cronobacter sakazakii*, EPEC, *Enterobacter cloacae*, *Salmonella pullorum*, and. *S. typhimurium*, and have shown both anti-adhesive and anti-invasive effects in the presence of GOS [93,94,95]. Studies using FOS, which contains fructose monosaccharide in its structure, against *E. coli* resulted in the inhibition of adhesion to human intestinal epithelial cells [96,97].

Apart from the anti-adhesive properties of HMOs, several studies have demonstrated the antibacterial activities of HMOs against different pathogenic bacteria.

A large portion of protein in milk that imparts a significant barrier to pathogen invasion in the gut is N-glycans. The milk oligosaccharides attached to the protein called N-linked glycans are more active than the free oligosaccharides. Yue et al. (2020) reported the MIC values for N-linked and free milk oligosaccharides against *S. aureus* as 256 nmol/mL and 32,768 nmol/mL, respectively. This implies that N-glycans are more active than free milk oligosaccharides [98]. The antibacterial activity of N-glycans is comparable with commercially available antibiotics, such as kanamycin. The growth inhibition of *S. aureus* and *S. typhimurium* by N-glycans exceeded the activity of kanamycin, whereas a moderate inhibition was observed against *E. coli* and *L. monocytogenes* [99]. Another study, however, showed a notable decline in the growth of *E. coli, Peptostreptococcaceae,* and *C. jejuni* in the presence of LNnT. Interestingly, LNnT also exhibits stronger antibacterial activity against GBS, *S. agalactiae* strain GB590, and *S. agalactiae* strain GB2 [100]. The fucose moiety of N-glycans is responsible for the antipathogenic effect as defucosylation of N-glycans resulted in lowered or negligible inhibition for all the pathogens [101]. Furthermore, the growth inhibitory effect of neutral HMOs is also reported against *S. agalactiae* (GBS) and its different serotypes. Growth reduction above ~95% was observed in GBS serotype III, Ia, and V [91]. One of the studies demonstrated that neutral HMOs from different donors decreased *S. agalactiae* GB590 growth by 80% and *S. agalactiae* GB2 growth by 95%. The same study also showed a reduction in *S. agalactiae* strain GB2 viability by 36% [98]. In particular, 2′-FL reduced the growth of *S. agalactiae* strain GB590 by 20% within 4 h and 11% after 24 h [89]. In another study, 2′-FL successfully prevented the invasion of *Campylobacter jejuni, E. coli,* and *Peptostreptococcaceae*, thus retarding their growth and colonization [35]. Growth of *Deferribacteres, Anaerotruncus, Parabacteroides, Eubacterium, Mucispirillum, Patescibacteria,* and *Alistipes* was sharply suppressed in the presence of 2′-FL [57]. Acidic HMOs such as 6′-SL effectively inhibit *P. aeruginosa*, a crucial opportunistic pathogen that leads to detrimental infections, such as infection of human pneumocytes in cystic fibrosis. HMOs enter the systemic circulation from the gut and then reach the respiratory system where they modulate the pathogen–host interaction by binding to the pathogen and not to the respiratory cells [101].

### 4.3. Antibiofilm Activity of HMOs

Considering the immense benefits of HMOs and their diverse antibacterial properties, researchers have evaluated the antibiofilm potential of HMO molecules. The presence of HMOs has shown inhibitory effects against biofilm assembly. Up to 93% biofilm inhibition by HMOs was observed against the *S. agalactiae* strain and up to 60% inhibition was observed against methicillin-resistant *S. aureus* (MRSA) [102]. A recent study demonstrated that GB2 had pronounced vulnerability towards HMOs [103]. The same study also showed that HMOs decreased biofilm production in *S. aureus* by 30–60% [104]. An *in vitro* study showed that the presence of HMOs caused a significant decline in the formation of biofilms in multi- and pan-drug-resistant *A. baumanii*. HMOs mainly act by suppressing the formation of pellicles (floating biofilms). The same study also revealed that HMOs potently inhibited biofilm establishment but were not effective in distorting pre-existing biofilms [105]. The biofilm matrix aids the dilution of antibiotic molecules either by lowering their diffusion into the biofilm or by other machinery. This calls for novel techniques that disrupt these mechanisms and facilitate the action of antibiotics. HMOs are shown to potentiate the action of several antibiotics such as clindamycin, erythromycin, gentamicin, and minocycline against GBS [106]. HMOs with varying monosaccharide constituents have shown a greater capability in inhibiting biofilm formation. Neutral HMOs, specifically 2’-FL, lack efficient antibiofilm activity. However, the conversion of 2′-FL to an anionic, amino derivative illustrated significant results [107]. The alteration of 2′-FL by amination (Kochetkov amination) forming the cationic molecule 1-amino-2′-fucosyllactose showed biofilm inhibitory activity. The mechanism of biofilm inhibition was not clearly understood but it was hypothesized that the inhibitory action was due to the interactions between the cationic fucosyllactose entity with the anionic EPS matrix and the negatively charged DNA structure, which led to the inclusion of a positive charge in the biofilm structure. The introduction of cationic charges in the biofilm structure thus dismantled the biofilm. Therefore, the use of 2′-FL derivatives executed biofilm inhibitory activity against *S. agalactiae* GB2 and GB590 and reduced the biofilm production by 46% in GB2 and 37% in GB590 [108]. The antibacterial and antibiofilm activities of different HMOs are listed in Table 2. In a recent study by Sylwia Jarzynka et al., the antibiofilm activity of total and fractionated HMO molecules were evaluated against mature biofilms of different clinically relevant Gram-negative and Gram-positive bacteria. They observed isolate and strain-specific reductions in the number of biofilm cells among Gram-positive bacteria, specifically in *E. faecalis* and *S. aureus*. However, HMOs did not show any significant activity against pre-formed biofilms of Gram-negative bacteria [109].

Summarizing all the results, we can conclude that HMO molecules can modulate the initial stages of biofilm formation and prevent bacterial colonization. However, the underlying molecular mechanism is not yet well understood. Researchers hypothesize that the antibiofilm activity of HMOs may be due to their structural resemblance with the bacterial polysaccharides, that in turn modulate the gene expression profile of bacterial EPS, needed to maintain a structured biofilm community [104,113,114]. Focused *in vivo* studies in this context are, therefore, necessary to prove this hypothesis. 

## 5. Conclusions and Future Directions

Viral infections and chronic biofilm infections present a difficult challenge for humanity, calling for a strong need to develop anti-viral [115] and antibiofilm therapies [116]. The unconjugated and structurally complex glycans, HMOs, have emerged as integral components of the human milk glycome, shaping the immunity and microbial communities of the infant’s gut. The HMOs confer protection to the infant from microbial diseases by promoting the growth of commensal bacteria, such as *Bifidobacteria*, *Akkermansia*, and *Lactobacillus*, by directly killing pathogenic bacteria, such as GBS, by conferring anti-adhesive properties of ‘decoy’ receptors against enteropathogens and by reducing biofilm formation of bacterial pathogens such as *A. baumanii*, *S. aureus* and *S. agalactiae.* The adhesion of HMOs on the tissues such as epithelial cells or urinary bladder cells confers protection to the host lining from pathogens that cause chronic infections, e.g., pooled HMOs have been used to reduce *in vivo* vaginal GBS colonization. Antimicrobial therapy is ineffective in treating chronic infections because of the rapid horizontal transfer of antimicrobial resistance genes, antibiotic tolerance, and the presence of the persister phenotype inside the biofilm matrix, thereby causing a relapse of the infection. In the pipeline of non-antibiotic-based antibiofilm therapeutics, HMOs have shown significant promise in inhibiting biofilms of clinically relevant pathogens such as GBS, *S. aureus*, and *S. agalactiae*. 

The interdisciplinary interactions among clinicians, microbiologists, and chemists can tap into several unexplored aspects of HMO research, such as the interactions of HMOs with capsular polysaccharides, effects on cell elongation and division of bacteria, permeability changes in the bacterial membrane, expression of bacterial adhesins in the presence of HMOs, interactions of lectin-based adhesins (FimH type 1 pilus lectin) with HMO-glycans, the ability of HMOs to jam quorum-sensing signaling, restoring the susceptibility of XDR bacteria to resistant drugs, and developing adjuvant therapies using HMOs to treat biofilms in chronic infections. HMOs are undigested by host digestive enzymes, stay intact in the gut for longer durations, and, thus, play a vital role in impacting the gut microflora and immunity. There is a strong need for more research effort to understand the untapped, broad application of HMOs to biofilm infections in other tissues of the body other than the vaginal mucosa and intestines. The anti-adhesive, antibacterial, and antibiofilm characteristics of HMOs make them excellent biomolecular entities that can be modified using biocidal moieties to develop antibiofilm agents, with the additional advantage of not adversely affecting the essential gut microbiome. It is extremely important to leverage the latest computational technologies and most relevant animal models to accelerate the discovery of novel and effective antibiofilm agents. 

Recent advancements in glycan-focused machine learning have led to the development of unique tools used to study HMO-mediated host–microbe interactions. Glycobiology modifications in HMOs that can result in lowered immunogenicity to the host can be predicted in further research by using a deep-learning-based, SweetTalk, glycans language model. Computational biologists have struggled to incorporate diverse carbohydrate structures in their workflows, which can now be easily done using Glycowork (a python package for glycan data science and machine learning); therefore, they can now study in silico interactions of modified HMOs with the host tissues and the pathogens. ML approaches such as logistic regression, random forest, support vector machines (SVMs), and neural networks are increasingly being used to discover new antibiofilm agents. It employs a training set that consists of small molecules or peptides with information about their experimentally known biofilm inhibiting/killing activity and their chemical properties (generated using the QSAR chemoinformatics approach) to train an algorithm. The algorithm is trained in such a way that it creates a mathematical relationship between the antibiofilm activity of the molecule and the features of each molecule. This trained model is used to check the antibiofilm properties of known antibiofilm agents and also of unknown antibiofilm agents, which is validated later using *in vitro* studies. Unlike antibiofilm glycans or HMOs, antimicrobial peptides with antibiofilm properties have been predicted through numerous ML approaches. A database of antibiofilm HMOs as a result of coordinated in silico studies using advanced approaches can definitely provide interesting *in vitro* and *in vivo* antibiofilm leads. Although *in vitro* tools enable high-throughput screening under controlled conditions, they give a limited understanding of the impact of host proteins, immune responses, stress factors, and the complex chemical and physical environment that microbes adapt to. It is also essential to validate the promising *in vitro* antibiofilm strategies *in vivo* for benchside-to-bedside translation using appropriate animal models representing the appropriate host milieu conditions. 

A major challenge in HMO studies is the inadequate availability of HMO molecules. The synthesized HMOs are expensive and only represent a negligible proportion of the whole human milk glycome. More than 200 unique branched structured HMO molecules are found in human milk, of which synthetic HMOs are restricted to only structurally small and simple HMOs such as trisaccharide and tetrasaccharides. Pooled milk from donors is another option available for studying HMO molecules. The main limitation of pooled milk is the variation in HMO composition and difference in activity in the absence of the maternal phenotype. The HMOs composition in milk changes regularly, perhaps to boost immunity. However, the isolation of the exhausted HMO molecule responsible for biological activity is difficult in a pooled sample. Hence, innovative solutions are required to identify the HMO molecules with antimicrobial activity that would further assist in the chemoenzymatic synthesis of HMO molecules. Such initiatives will certainly boost multidisciplinary research on the human glycome and assist in the development of novel therapeutic or prophylactic HMO cocktails against infections.

## Figures and Tables

**Figure 1 nutrients-14-05112-f001:**
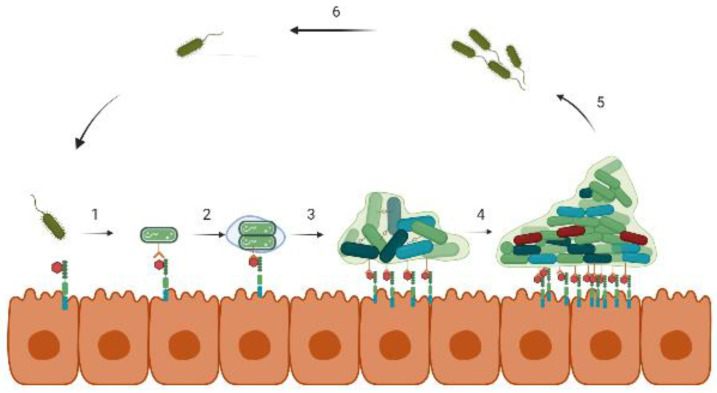
Stages of biofilm formation on a biotic surface. (1) Bacteria approach the host cells and bind to specific receptor proteins. (2) After binding, bacteria start multiplying and secrets EPS that (3) eventually forms the microcolonies, and (4) the biofilm develops into a more complex 3D structure. In (5) and (6), the cycle, thus, continues with the dispersion of bacteria from the biofilm.

**Figure 2 nutrients-14-05112-f002:**
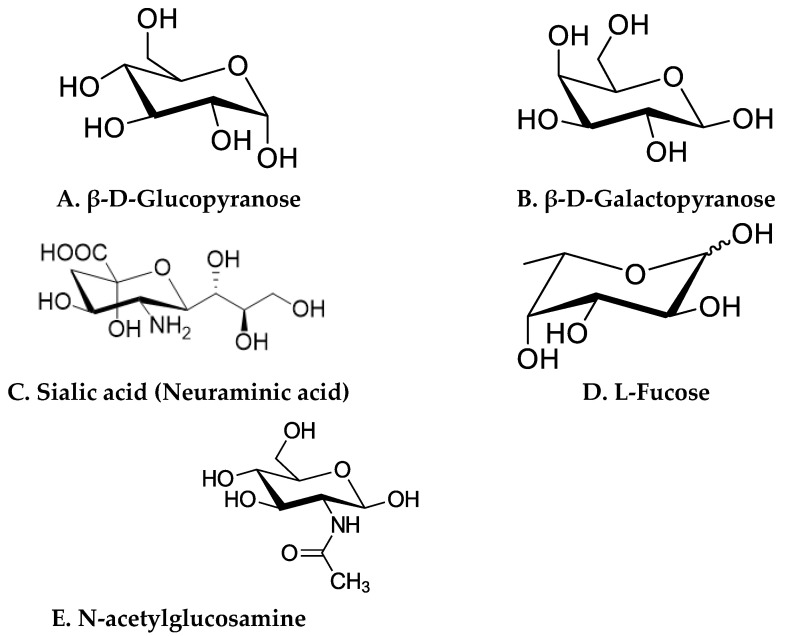
Structures of basic monosaccharide constituents of HMOs. (**A**) β-D-Glucopyranose, (**B**) β-D-Galactopyranose, (**C**) Sialic acid (Neuraminic acid), (**D**) L-Fucose, (**E**) N-acetylglucosamine.

**Figure 3 nutrients-14-05112-f003:**
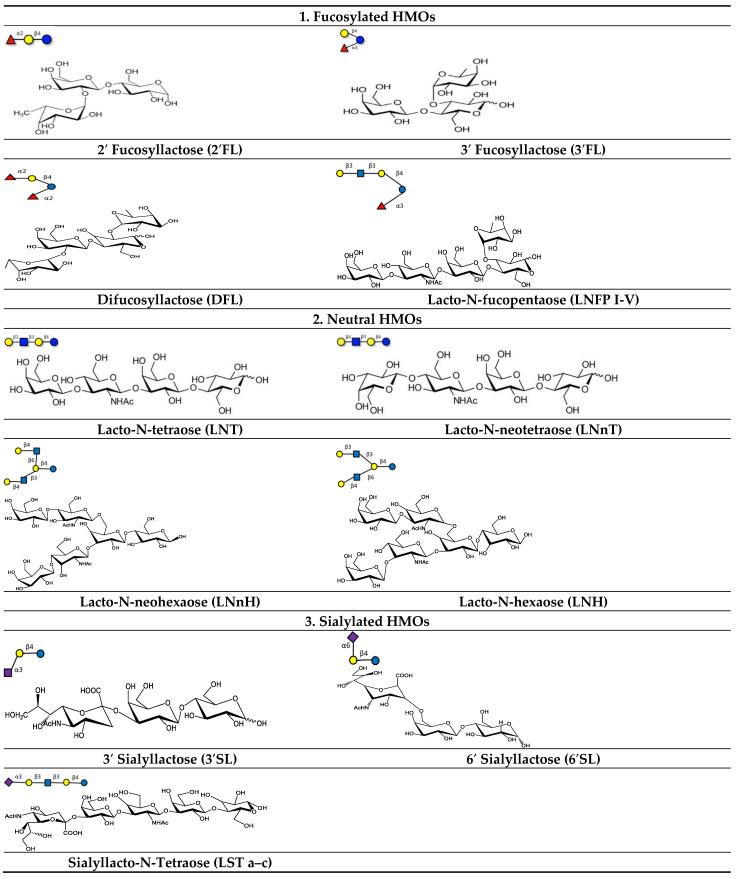
Different types of fucosylated, neutral, and sialylated HMO molecules.

**Figure 4 nutrients-14-05112-f004:**
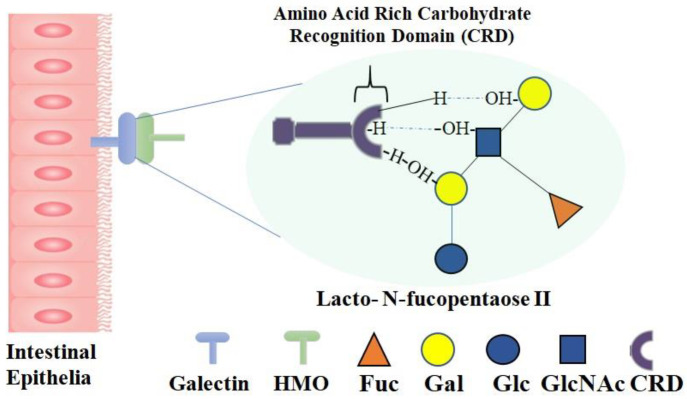
Galectin–HMO Binding. Human milk oligosaccharides bind to galectin expressed by intestinal epithelia; zoom-in image of gelectin-HMO binding site shows the detailed molecular representation of chemical interactions: OH groups of galactose and N-acetylglucosamine in HMOs form hydrogen bonds with amino acid residues (e.g., His44, Arg48, Trp68, Val59, Asn61, Asn46, Glu71, and Arg73) in the CRD region of galectin.

**Figure 5 nutrients-14-05112-f005:**
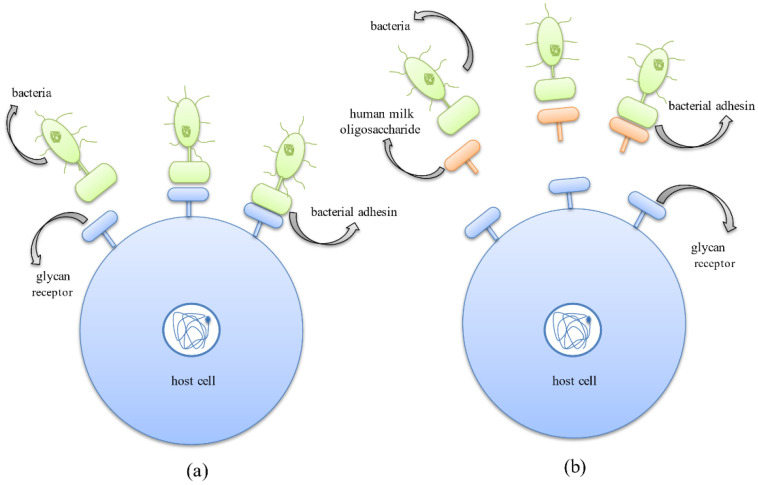
Anti-adhesive mechanism exhibited by HMOs where they behave as soluble decoy receptors, inhibiting bacterial adhesion to host cell surface receptors. (**a**) Binding of bacterial protein adhesins to the host cell surface glycan receptors, aiding successful colonization in the host system. (**b**) Binding of bacterial protein adhesins to HMOs thus inhibiting adhesion to host cells and successfully preventing the occurrence of infection.

**Figure 6 nutrients-14-05112-f006:**
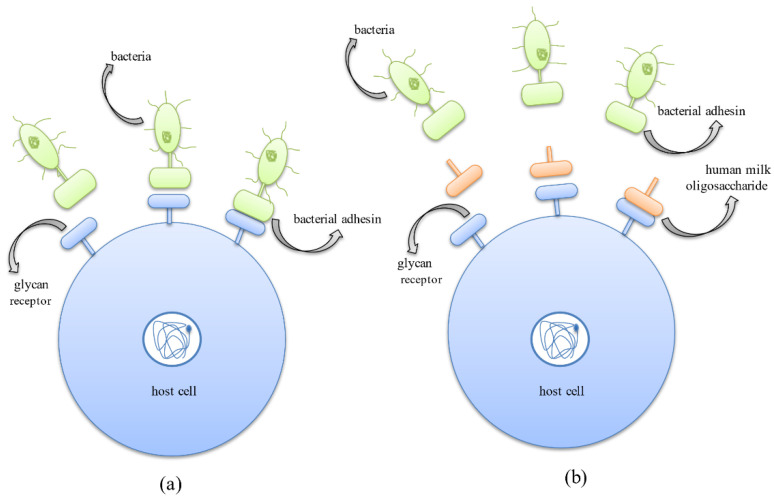
Anti-adhesive mechanism exhibited by human milk oligosaccharides where they compete with bacterial protein adhesins for binding to host cell surface glycan receptors. (**a**) Binding of bacterial protein adhesins to the host cell surface glycan receptors aiding successful colonization in the host system. (**b**) Binding of HMOs to host cell surface receptors thus inhibiting adhesion of bacteria to host cells and successfully preventing the occurrence of infection.

**Table 1 nutrients-14-05112-t001:** Major tissue and device-associated biofilm infections.

	Biofilm Infection	Associated Pathogens	References
Tissue-associated infections	Chronic rhinosinusitis	*Corynebacterium tuberculostearicum*, *Haemophilus influenzae*, *Lactobacillus sakei*, *P. aeruginosa*, *S. aureus*, *Streptococcus pneumoniae*	[11,12]
	Periodontitis	*Fusobacterium nucleatum*, *Porphyromonus gingivalis*, *Tannerella forsythia*, *Treponema denticola*,	[1,11]
	Pharyngitis:	*Group A Streptococcus (GAS)*, *H. influenzae*, *S. aureus*	
	Otitis media	*GAS*, *H. influenzae*, *P. aeruginosa*, *S. pneumoniae*	
	Infective endocarditis	*Lactobacillus lactis*, *S. aureus*	
	Cystic fibrosis	*Burkholderia cenocepacia*, *H. influenza*, *P. aeruginosa*, *S. aureus*	
	Colorectal cancer and ulcerative colitis	*Bacteriodes fragilis*, *Enterobacteriaceae*, *E. coli Fusobacterium* spp., *Shigella* spp.	
	Vaginosis	*Bacteroides*, *Gardnerella vaginalis*, *Mycoplasma*	
	Urinary tract infections	*E. coli*, *Enterobacter* spp., *Klebsiella pneumonia*, *Proteus* spp., *Staphylococcus saprophyticus*	
	Prostatitis	*E. coli*	
	Osteomyelitis	*E. coli*, *H. influenzae*, *Streptococcus agalactiae*, *S. aureus*, *Streptococcus pyogenes*	
	Wound infections	*P. aeruginosa*, *S. aureus*	
Device-associated Infections	Contact lenses	*P. aeruginosa*, *S. aureus*	[13]
	Dental implants	*Aggregatibacter actinomycetemcomitans*, *Eikenella corrodens*, *P. gingivalis,*	[14]
	Endotracheal tubes	*Acinetobacter baumannii, Enterobacter* spp., *Enterococcus faecalis*, *K. pneumoniae*, *P. aeruginosa*, *S. aureus*	[15]
	Prosthetic joints	*P. aeruginosa*, *S. aureus*, *S. epidermidis*	[16]
	Vascular catheters	*K. pneumonia*, *P. aeruginosa*, *Coagulase-negative Staphylococci*, *S. aureus*	[17]
	Vascular grafts	*P. aeruginosa*, *S. aureus*	[18]

**Table 2 nutrients-14-05112-t002:** Antibacterial and antibiofilm activity of major HMO molecules.

HMOs	Pathogens	Antibacterial Properties	References
2′-Fucosyllactose (2′-FL)	*C. jejuni,* Enteropathogenic *E. coli, P. aeruginosa, S. enterica* serovar *fyris, S. dysenteriae,*	Interferes with binding of specific receptors on epithelial cells and hence prevents infection development	[90,108,110,111]
	*A. baumanii*	Represses biofilm formation	[105]
3-Fucosyllactose (3′-FL)	*C. jejuni, E. coli, P. aeruginosa*	Prevents adhesion of bacteria to receptors	[88,90,108]
	Group B *Streptococcus*	10% decrease in biofilm formation	[90]
	*A. baumanii*	Represses biofilm formation	[105]
3′-sialyllactose (3′-SL)	*E. coli, H. pylori, S. fyris, V. cholerae*	Inhibits bacterial adhesion	[87,88,112]
	GBS	10% decrease in biofilm biomass	[102]
6′-sialyllactose (6′-SL)	*E. coli, S. fyris, V. cholerae*	Interferes with bacterial adhesion	[87]
	GBS	9% decrease in biofilm biomass	[102]
1-amino-2′-FL	GBS 590	37% decrease in biofilm production	[89]
	GBS 2	46% decrease in biofilm production	
Lacto-N-neotetraose (LNnT)	GBS	13% decrease in biofilm biomass	[102]
	*A. baumanii*	Represses biofilm formation	[103]
Difucosyllactose (DFL)	*A. baumanii*	Represses biofilm formation	
Lacto-N-fucopentaose I (LNFP I)	*A. baumanii*	Represses biofilm formation	
Lacto-N-fucopentaose II (LNFP II)	*A. baumanii*	Represses biofilm formation	
Lacto-N-fucopentaose III (LNFP III)	*A. baumanii*	Represses biofilm formation	
Lacto-N-triose II (LNT II)	*A. baumanii*	Represses biofilm formation	

## Data Availability

Not applicable.

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
