# Peer review of "Human Milk Oligosaccharides as Potential Antibiofilm Agents: A Review"

_nutrients, 2022, doi:10.3390/nu14235112_

Round 1

Reviewer 1 Report

Dear authors,

I really appreciate this complete review of the HMOs.

Please check the line 170 the reference of Moro et al is missing. I suggest to add this reference and review the numbering 

Author Response

Point1. Please check the line 170 the reference of Moro et al is missing. I suggest to add this reference and review the numbering
Response 1: The authors would like to thank the reviewer for the suggestions. We have cited the observations of Moro with the appropriate citation in line no. 107 (We would like to bring to the notice of the reviewer that Moro et al., was missing at line 107 and not Line 170)
We have incorporated it as Ref 39 in the manuscript. Now line 111.
In addition to this, we have done minor English corrections
The authors appreciate reviewer for thorough reading and valuable inputs for overall improvement of the manuscript.

Reviewer 2 Report

Bhowmik et al. and co-workers summarized the potential of HMOs as antibacterial and anti-biofilm agents and proposed further investigations on using HMOs for new therapeutic interventions. This review is well written, gives a very comprehensive presentation of HMOs, and is constructive for understanding the current state and the need for future studies. Thus, I think the manuscript is suitable for publication in Nutrients journal, and I suggest a minor revision as follows:

In section 3, a molecular graphic that shows HMOs binding to the receptor would be helpful to improve intelligibility, especially a zoom-in figure for the binding situation in the carbohydrate-recognition domain.

Author Response

Point 1. In section 3, a molecular graphic that shows HMOs binding to the receptor would be helpful to improve intelligibility, especially a zoom-in figure for the binding situation in the carbohydrate-recognition domain.
Response 1. The authors would like to thank the reviewer for the suggestions. We have included a molecular graphic (in Figure 3 of revised version) showing the binding situation in the carbohydraterecognition domain. Also, the authors would like to appreciate critical and careful review of the manuscript by the reviewers.
We have incorporated an additional figure (Fig 3) under section 3.
In addition to this, we have done minor English corrections.
The authors appreciate reviewer for a careful and diligent review of our manuscript and sharing valuable feedback for overall improvement of the manuscript.